# The Current Landscape of Glioblastoma Biomarkers in Body Fluids

**DOI:** 10.3390/cancers15153804

**Published:** 2023-07-26

**Authors:** Saba Zanganeh, Elham Abbasgholinejad, Mohammad Doroudian, Nazanin Esmaelizad, Fatemeh Farjadian, Soumya Rahima Benhabbour

**Affiliations:** 1Joint Department of Biomedical Engineering, North Carolina State University and The University of North Carolina at Chapel Hill, Chapel Hill, NC 27599, USA; sabaz@unc.edu; 2Department of Cell and Molecular Sciences, Faculty of Biological Sciences, Kharazmi University, Tehran 15719-14911, Iran; elham79agn@gmail.com (E.A.); nesmaelizad@gmail.com (N.E.); 3Pharmaceutical Sciences Research Center, Shiraz University of Medical Sciences, Shiraz 71348-14336, Iran; fatemehfarjadian@gmail.com; 4Division of Pharmacoengineering and Molecular Pharmaceutics, UNC Eshelman School of Pharmacy, University of North Carolina at Chapel Hill, Chapel Hill, NC 27599, USA

**Keywords:** glioblastoma, biomarkers, body fluids, circulating tumor cells (CTCs), cell-free DNA (cfDNA), cell-free RNA (cfRNA), microRNA (miRNA), extracellular vesicles (EV), exosomes

## Abstract

**Simple Summary:**

Glioblastoma is a highly aggressive brain cancer, and early detection and accurate diagnosis are crucial for effective treatment. Traditional diagnostic methods have limitations, and liquid biopsies offer a non-invasive and dynamic approach to detecting and monitoring glioblastoma. This review provides a comprehensive overview of various cancer biomarkers, including circulating tumor cells, cell-free DNA, and RNA, such as microRNA, as well as extracellular vesicles. It highlights their clinical utility in glioblastoma detection, monitoring, and prognosis. In addition, challenges and limitations in implementing liquid biopsy strategies in clinical practice are addressed.

**Abstract:**

Glioblastoma (GBM) is a highly aggressive and lethal primary brain cancer that necessitates early detection and accurate diagnosis for effective treatment and improved patient outcomes. Traditional diagnostic methods, such as imaging techniques and tissue biopsies, have limitations in providing real-time information and distinguishing treatment-related changes from tumor progression. Liquid biopsies, used to analyze biomarkers in body fluids, offer a non-invasive and dynamic approach to detecting and monitoring GBM. This article provides an overview of GBM biomarkers in body fluids, including circulating tumor cells (CTCs), cell-free DNA (cfDNA), cell-free RNA (cfRNA), microRNA (miRNA), and extracellular vesicles. It explores the clinical utility of these biomarkers for GBM detection, monitoring, and prognosis. Challenges and limitations in implementing liquid biopsy strategies in clinical practice are also discussed. The article highlights the potential of liquid biopsies as valuable tools for personalized GBM management but underscores the need for standardized protocols and further research to optimize their clinical utility.

## 1. Introduction

Glioblastoma (GBM) is the most prevalent and aggressive primary brain cancer in adults and is associated with poor survival and high mortality. GBM accounts for approximately 15.6% of brain tumors and 45.2% of primary malignant brain tumors in adults [1,2]. GBM is characterized by its rapid growth, invasive nature, and resistance to standard therapeutic modalities such as surgery, radiation, and chemotherapy. The average survival rate after diagnosis is a mere 12–15 months, rendering it an extremely aggressive tumor with a less than 5% overall 5-year survival rate [3,4,5].

Early detection and accurate diagnosis of glioblastoma are of paramount importance for improving patient outcomes, as they facilitate the timely implementation of appropriate treatment strategies and more effective disease management. Delays in diagnosis can lead to rapid disease progression due to the aggressive nature of glioblastoma. Moreover, early detection enables differentiation between glioblastoma and other brain tumors or non-neoplastic conditions, thereby enabling personalized treatment plans [6,7].

Currently, tissue biopsies and imaging methods are used to diagnose GBM [8]. Imaging techniques are unable to reliably distinguish between lesions brought on by true tumor growth and pseudoprogression, which are treatment-related lesions that mimic tumor advancement and may eventually go away on their own. This may result in delayed therapeutic interventions and an incorrect prognosis of the response to therapy. Similar to this, tissue biopsies require invasive techniques and only offer a static image of a tumor that is continually changing. To monitor real-time tumor dynamics during treatment, the collection of regular tissue samples is not possible [9]. In contrast, the advantage of non-invasive liquid biopsies, which enable the detection of circulating biomarkers, is the ease of repeated sampling and the ability to track any dynamic changes in the tumor during therapy [6].

For the identification, diagnosis, and follow-up of glioblastoma, biomarkers found in body fluids, such as cerebrospinal fluid (CSF), blood, and urine, can provide important information. Several biological analytes can be found in body fluids and obtained through liquid biopsies, including circulating tumor cells (CTCs), circulating cell-free DNA (cfDNA) containing circulating tumor DNA (ctDNA), circulating cell-free tumor RNA (ctRNA), containing messenger RNA (mRNA), and small RNAs, extracellular vesicles (EVs) [10,11,12]. These circulating biomarkers can originate from the tumor tissue and, as such, can act as a representative sample of glioblastoma. Analysis of these biomarkers, also known as liquid biopsies, provides a non-invasive and real-time approach to assessing tumor characteristics, monitoring treatment response, and detecting disease recurrence [13,14]. These biomarkers have been involved in ongoing clinical studies using body fluids in GBM patients (Table 1).

Various liquid biopsy tests have been granted approval by the U.S. Food and Drug Administration (FDA) to be used in different solid cancers. Notably, Guardant360 is one such liquid biopsy test employed to analyze ctDNA from a simple blood draw. It serves to detect specific genetic alterations in tumor cells of patients with lung cancer, breast cancer, colorectal cancer, and prostate cancer [15]. Similarly, FoundationOne Liquid, another FDA-approved comprehensive liquid biopsy test, can analyze ctDNA and circulating tumor RNA (ctRNA) from blood samples, detecting genomic alterations, including mutations, copy number variations, and gene rearrangements [16]. Additionally, the Cobas EGFR Mutation Test v2 is an FDA-approved liquid biopsy test designed for non-small cell lung cancer (NSCLC) patients, focusing on identifying mutations in the epidermal growth factor receptor (EGFR) gene [17]. Moreover, the Epi proColon liquid biopsy test is FDA-approved for colorectal cancer screening, serving to detect methylated Septin 9 (SEPT9) DNA in blood samples [18]. However, no liquid biopsy test has received approval for GBM. GBM presents several challenges for the application of liquid biopsy tests compared to other solid cancers. First, GBM is a primarily brain-based tumor, and the release of ctDNA into the bloodstream is relatively limited compared to tumors located in other organs, primarily due to BBB. As a result, the levels of ctDNA in the blood of GBM patients are often low, making it more challenging to detect and analyze tumor-specific mutations [19]. Furthermore, GBM is known for its genetic heterogeneity, meaning different regions of the tumor may have distinct genetic alterations. Liquid biopsies might not fully capture the entire genetic landscape of the tumor due to the limited ctDNA shedding and the challenge of sampling different tumor regions effectively. Additionally, GBM shares some genetic alterations with other brain tumors and non-tumor-related neurological conditions. Distinguishing ctDNA derived from GBM from that of other brain-related sources can be complex, potentially leading to false-positive or false-negative results [20]. Due to these challenges, the application of liquid biopsies, including tests like Guardant360, in GBM remains an area of ongoing research and development. While liquid biopsies have shown great promise in other solid cancers, their implementation in GBM diagnosis and monitoring requires further investigation to overcome the unique obstacles presented by this aggressive brain tumor [21].

This article provides a comprehensive overview of glioblastoma biomarkers in body fluids, encompassing cell-free DNA, RNA, and miRNA, circulating tumor cells, and extracellular vesicles such as exosomes and microvesicles. The discussion also explores the current clinical utilization of biomarkers in glioblastoma detection and monitoring, as well as the challenges and limitations associated with implementing these strategies in clinical practice.

## 2. Circulating Tumor Cells as Glioblastoma Biomarkers

CTCs refer to malignant cells that have detached from the primary tumor and entered the bloodstream or lymphatic system. These cells offer valuable insights into the genetic and phenotypic characteristics of the tumor, serving as potential indicators of metastatic capability. CTCs are relatively scarce compared to other blood cells, and their presence in bodily fluids can vary significantly depending on the stage and type of tumor [22,23].

CTCs display three identifiable characteristics: they originate from the primary tumor or metastases, are found within blood vessels, and actively participate in the circulatory system. The presence of CTCs poses significant risks, as they play a critical role in the distant spread of cancer and contribute to the mortality associated with malignant tumors. These cells exhibit various advanced properties, including epithelial-mesenchymal transition (EMT) and dormancy, which are crucial for their survival in the bloodstream. Moreover, CTCs possess specific attributes such as resistance to radiation and chemotherapy, evasion of anti-cancer immune responses, and the capacity to metastasize. CTCs originating from primary tumors or metastatic sites and persisting in the bloodstream are considered the most aggressive subset of cancer cells. Precise molecular characterization of these cells can provide valuable insights for clinicians, assisting in the detection of metastasis or recurrence, tracking disease advancement, and assessing treatment efficacy. However, it is worth noting that the majority of CTCs are quickly eliminated from circulation, suggesting that their capacity for metastasis may be restricted [24]. CTCs are released into bodily fluids through processes such as intravasation (entry into the bloodstream) and extravasation (exit from the bloodstream). Several techniques can be employed to detect CTCs, including immunocytochemistry (labeling specific cell surface markers), flow cytometry (sorting cells based on size and surface markers), and microfluidic devices (isolating CTCs based on size or specific antibody affinity). These methodologies aim to capture and quantify CTCs while characterizing their molecular and phenotypic properties [22].

Common methods utilized for detecting CTCs in GBM often involve isolating them from red blood cells through gradient centrifugation, either alone or followed by targeted removal of white blood cells using CD45. Subsequently, immunocytochemical staining is performed using GBM-specific markers, primarily glial fibrillary acidic protein (GFAP), or fluorescence in situ hybridization (FISH) to probe for aneuploidy, specifically targeting the chromosome 8 centromere. An alternative approach involves introducing green fluorescent protein under the control of the telomerase (hTERT) promoter via viral transduction to identify cancer cells for CTC detection [25]. CTCs were found in the blood of GBM patients using a telomerase promoter-based test by Macarthur et al. [26]. Similarly, Müller et al. used antibodies against GFAP and amplification of the epithelial growth factor receptor (EGFR) gene to find CTCs in the peripheral blood of GBM patients [27]. Sullivan et al. also used the CTC-iChip^®^ microfluidic technology in another study to find CTCs that were specific to glioblastoma [28,29]. Several technologies have been developed for the identification and characterization of CTCs, including immunomagnetic cell enrichment, flow cytometry-based systems, as well as automated microscopy systems. However, only one technology, CellSearch^®^ (Janssen Diagnostics, Raritan, NJ, USA), has obtained approval from the U.S. FDA for the detection of CTCs and monitoring disease progression in metastatic breast, prostate, and colorectal cancers, based on clinical trials. This technology primarily relies on the detection of epithelial cell adhesion molecules (EpCAM), which may limit its effectiveness in detecting CTCs derived from primary brain tumors that may not express such surface markers. The presence of CTCs in GBM has been a topic of discussion for many years. GBM is known for its high aggressiveness and invasiveness, but it predominantly remains localized to the brain, with extracranial metastases being extremely rare, affecting only 0.5–2% of patients and typically involving sites like the lungs, bone, lymph nodes, or liver [30,31,32,33,34,35,36]. Mounting evidence has shown that the presence and quantity of CTCs in bodily fluids are associated with glioblastoma progression, treatment response, and patient survival, indicating their potential as diagnostic and prognostic biomarkers. For instance, higher CTC counts have been correlated with poorer prognosis and shorter overall survival in glioblastoma patients. Additionally, changes in CTC counts during treatment can provide insights into treatment efficacy and the likelihood of disease recurrence [6].

CTCs represent a comprehensive reflection of tumor properties, and even the analysis of single CTCs can yield genomic information. Detection of CTCs in patients with glioma has been achieved using telomerase assays, with successful identification in a subset of patients. Moreover, there have been cases where the quantity of CTCs has shown a correlation with the efficacy of radiation therapy. Patients who responded well to the treatment demonstrated a notable decrease in CTC count after undergoing therapy. In another investigation, isolated peripheral mononuclear cells were used to detect glioblastoma CTCs using GFAP. The results showed 20% of the 141 individuals in this study had CTCs, but in some cases, just one cell per patient was discovered in the 10 mL blood sample [13,26,27,28,37].

The use of CTCs as biomarkers faces several challenges. Current methods for CTC detection may not be optimal, as they require extensive microscopic screening to find GBM markers in millions of cells and because negative enrichment techniques may cause the loss of already rare CTCs due to possible antibody cross-reactivity or physical co-segregation of CTCs with blood cells. Additionally, the current GBM-CTC detection markers lack specificity, making it difficult to separate CTCs from lingering uncommon or regular blood cells. This limitation can result in elevated background staining, false-positive detections, or reduced sensitivity due to variations in marker expression among GBM cells, leading to the potential omission of certain CTCs. Standardization of CTC isolation and characterization protocols is also necessary to ensure the reproducibility and comparability of results across different studies and clinical settings [25].

## 3. Cell-Free Nucleic Acids as Glioblastoma Biomarkers

The cfNAs encompass fragments of DNA, RNA, and miRNA that are liberated into bodily fluids by various cells, including tumor cells. These nucleic acids offer valuable insights into the genetic and epigenetic alterations present in glioblastoma. cfDNA, which consists of double-stranded DNA derived from both normal and tumor cells, and Cell-Free RNA (cfRNA), including mRNA and non-coding RNA such as miRNA involved in gene regulation, represent the constituents of cfNAs [22,38].

The release of cfNAs into bodily fluids occurs through processes such as apoptosis, necrosis, and active secretion by living cells. Various molecular techniques, including next-generation sequencing (NGS), polymerase chain reaction (PCR), and droplet digital PCR (ddPCR), enable the detection of cfNAs. These techniques facilitate the amplification and analysis of specific nucleic acid sequences, allowing for the identification of tumor-specific mutations, gene expression patterns, and miRNA profiles [6,39,40]. Numerous studies have demonstrated the potential of cfNAs as diagnostic and prognostic biomarkers in glioblastoma [22]. The unique epigenetic signatures present in cfDNA can differentiate between malignancies with similar origins but distinct clinical trajectories [41].

### 3.1. Cell-Free DNA

The detection of circulating tumor DNA (ctDNA) in glioblastoma faces challenges due to the lower levels of cell-free DNA (cfDNA) compared to other solid tumors. Nabavizadeh and colleagues conducted a study that investigated the relationship between metrics related to blood-brain barrier (BBB) disruption, tumor-associated macrophages, and the concentration of plasma cfDNA and ctDNA in glioblastoma patients. Their findings revealed a positive correlation between the concentration of plasma cfDNA and the volume of the tumor exhibiting increased K_trans_ and K_ep_. K_trans_ represents the transfer of substances between plasma and the surrounding extracellular space, indicating BBB permeability, while K_ep_ reflects the rate at which contrast agent washes out from the extracellular space back into the bloodstream. The study also demonstrated a connection between higher tumor macrophage density, larger tumor vessel size, and elevated levels of plasma cfDNA. More specifically, a higher density of CD68+ macrophages surrounding blood vessels was associated with lower BBB permeability and a decreased detection rate of somatic mutations in plasma samples from glioblastoma patients [42].

In a preliminary investigation carried out by Bagley et al., a group of 42 individuals diagnosed with glioblastoma was enrolled to assess the potential clinical applications of plasma cfDNA. The primary objectives of the study were to identify somatic tumor mutations, evaluate the use of cfDNA as a prognostic indicator and measure of tumor burden, and aid in distinguishing true tumor progression from pseudoprogression. The researchers postulated that a high concentration of plasma cfDNA would be correlated with a poorer prognosis in newly diagnosed glioblastoma cases. To identify somatic tumor mutations, targeted NGS panels were employed to analyze plasma samples obtained before the initial surgical resection, enabling further molecular profiling. The study successfully demonstrated the feasibility of serial and longitudinal collection of plasma cfDNA in newly diagnosed glioblastoma patients, providing quantitative measurements. The findings indicated that plasma cfDNA could serve as a prognostic tool, a surrogate marker for tumor burden and progression, and a substrate for molecular profiling, including the detection of mutations through NGS analysis, which could aid in identifying both sensitivity and resistance to targeted therapies. Thus, plasma cfDNA shows promise as a valuable biomarker for the prognosis of individuals with newly diagnosed glioblastoma [43].

In another investigation, Fontanilles et al. evaluated the potential of cfDNA and ctDNA as biomarkers for disease monitoring in 49 glioblastoma patients. The study observed significant variations in cfDNA concentration during treatment, with a decrease after initial surgery or biopsy and an increase during disease progression in the temozolomide maintenance phase. The study indicated that cfDNA could serve as a prognostic marker for disease progression and survival in glioblastoma patients. However, further research is necessary to elucidate the prognostic significance of baseline cfDNA concentration and its kinetics before and after treatment [44].

A separate study explored the feasibility of cfDNA as a diagnostic tool for glioblastoma. The investigation demonstrated that measuring cfDNA concentration in glioblastoma patients is feasible and revealed a correlation between cfDNA levels and treatment response, with the highest levels observed during disease progression. These findings suggest the promising utility of cfDNA as a biomarker for monitoring tumor activity and response to treatment in glioblastoma patients [45].

A cohort of 62 patients diagnosed with isocitrate dehydrogenase (IDH) wild-type GBM was included in a study investigating the relationship between plasma cfDNA concentration and survival outcomes. The aim was to assess whether there is an association between pre-operative cfDNA levels and progression-free survival (PFS) as well as overall survival (OS). The results revealed that patients with high pre-operative cfDNA concentration had significantly poorer PFS and OS, even after adjusting for established prognostic factors. Additionally, the study observed that an increase in cfDNA concentration after undergoing chemoradiotherapy, compared to the baseline levels before surgery, was linked to worse PFS and OS. This noninvasive biomarker has the potential to enhance personalized treatment planning and aid in stratifying patients in clinical trials involving IDH wild-type glioblastoma [46].

A study conducted by Mouliere et al. proposed a non-invasive method for monitoring glioblastoma patients by detecting glioma-derived cfDNA in plasma and urine samples. This approach offers a more convenient and less restrictive monitoring option compared to CSF sampling. The study found that cfDNA fragments derived from tumors in CSF, plasma, and urine samples of glioma patients were shorter in length compared to non-mutant cfDNA. The researchers utilized personalized sequencing techniques to identify the size differences between mutant and non-mutant DNA in CSF, plasma, and urine samples. This personalized sequencing approach has the potential to revolutionize the detection and monitoring of glioblastoma by providing detailed genetic information, facilitating personalized treatment strategies, and enabling real-time monitoring of tumor dynamics [47].

GBM is known to be highly heterogeneous; therefore, it is important to understand the consistency of genomic profiling between primary tumors and other sources of tumor material. It has been found that ctDNA and CTCs were able to detect many of the same mutations as the primary tumor in GBM patients, but there were also some mutations that were only present in the primary tumor [6]. Another study indicated that ctDNA was able to detect mutations in GBM patients that were not present in the primary tumor, suggesting that ctDNA may provide additional information that is not captured by primary tumor biopsies [48]. However, the concordance between tissue and liquid biopsies may vary depending on the tumor entity, the amount and quality of cfDNA and CTCs, and the methods used for isolation and sequencing [49]. Overall, while there is promising evidence for the consistency of genomic profiling between primary tumors and cfDNA in GBM, concordance with CTCs remains an area that requires more comprehensive investigation [50].

DNA and RNA sequencing can provide valuable insights into the biology of cancer and inform the development of personalized treatment approaches. This can be a valuable tool for understanding tumor progression and identifying new mutations. The mutations driving tumor growth or contributing to drug resistance can be identified by sequencing the DNA or RNA of tumor cells. This information can be used to develop targeted therapies that are tailored to the specific mutations present in an individual’s tumor. In addition, sequencing can be used to monitor changes in the tumor over time, such as the emergence of new mutations or the development of drug resistance. This can help clinicians adapt their treatment strategies to better target the evolving tumor [51].

### 3.2. Cell-Free RNA

GBM is characterized by molecular diversity, making it difficult to develop effective diagnostic and treatment strategies. RNA molecules have shown promise as surrogate biomarkers for monitoring cancer progression and therapeutic responses. Various tumor-associated RNAs have been identified in peripheral blood, CSF, saliva, and urine. In GBM, RNA markers can be obtained from circulating cell-free RNA as well as from circulating exosomes, platelets, and CTCs [23,52,53,54].

Several molecular biomarkers have been identified in GBM that have prognostic and predictive value. These include mutations in the *IDH* gene, methylation of the O-6-methylguanine-DNA methyltransferase (*MGMT*) gene, mutations in the phosphatase and tensin homolog (*PTEN*) gene, mutation and/or amplification of the EGFR variant III (*EGFRvIII*), mutations in the *GFAP* gene, alterations in the telomerase reverse transcriptase (*TERT*) promoter, and loss of heterozygosity in chromosome 10. Epigenetic factors such as DNA methylation, histone modification, and microRNAs also play a role in regulating molecular processes and metabolic changes associated with GBM aggressiveness and recurrence. DNA methylation and microRNAs can impact metabolic processes including glycolysis, oxidative phosphorylation, and lipid metabolism through regulation of metabolism-related genes [21,55,56,57,58].

The discovery of cell-free miRNAs in body fluids, such as serum and plasma, has provided a non-invasive source of biomarkers for various diseases, including cancer. These miRNAs exhibit high stability and resistance to degradation, making them convenient biomarkers. They can also be easily sampled compared to tissue-based miRNAs [59,60]. Figure 1 illustrates the biogenesis process of miRNAs. In the context of GBM, a study indicated that miR-222 and miR-221 expression was significantly increased in GBM patients compared to healthy individuals, and elevated miRNA expression was associated with poorer survival outcomes [61]. Table 2 presents a list of miRNAs identified in body fluids that have been studied as potential biomarkers for GBM [62]. Differences in brain miRNA profiles may depend on the specific brain region, indicating that different types of brain tumors may be associated with distinct types and levels of miRNAs [12].

The advent of high-throughput technology has revealed that a significant portion of the genome, more than 90%, is transcribed into noncoding RNAs (ncRNAs), while only 2% encodes proteins. These noncoding RNAs, including small ncRNAs and long ncRNAs (lncRNAs), play crucial roles in various biological processes, primarily because most transcripts do not encode proteins. Among them, lncRNAs have been identified as active participants in biological pathways such as neural lineage commitment and immune response. Furthermore, lncRNAs have emerged as crucial regulators in disease processes, including tumor growth and metastasis in GBM [79].

In addition to ncRNAs, circulating cell-free mRNA (ccfmRNA) has been detected in the plasma of cancer patients and holds promise as a non-invasive biomarker for cancer diagnosis and monitoring. mRNA expression profiles can provide valuable information about the molecular characteristics of tumors, including their subtype, grade, and prognosis. A pilot study focusing on plasma ccfmRNA as a biomarker for glioma demonstrated that individuals with various glioma grades significantly differed in the expression of genes linked to malignant inflammation and immunity cross-talk. The transcriptome profile of plasma ccfmRNA and tumor samples, as well as with RNA information from The Cancer Genome Atlas (TCGA) glioma samples, were found to be positively correlated in the study. These findings imply that plasma ccfmRNA may be a non-invasive complement to radiological imaging for glioma diagnosis and surveillance [80].

Despite their potential, the use of cell-free nucleic acids as biomarkers faces several challenges and limitations. Cell-free nucleic acids are often present in low concentrations in body fluids, making their detection and analysis challenging. Highly sensitive detection techniques such as ddPCR and NGS can partially address this issue [6,14]. Furthermore, the presence of non-tumor cell-derived nucleic acids in body fluids can complicate the interpretation of the results. The heterogeneity of glioblastoma tumors can also lead to variations in the release and composition of cell-free nucleic acids, affecting their diagnostic and prognostic utility [6]. Additionally, the use of cell-free nucleic acids as biomarkers often requires advanced molecular techniques, such as NGS, which can be costly and time-consuming. Therefore, the development of more accessible and cost-effective detection methods is needed to facilitate the widespread adoption of cell-free nucleic acids as biomarkers for glioblastoma [1,22].

## 4. Extracellular Vesicles as Glioblastoma Biomarkers

Biofluids serve as easily accessible and effective sources for detecting glioblastoma through the identification of biomarkers. Glioblastoma and stromal cells release EVs that facilitate intercellular communication within the tumor microenvironment. EV-based liquid biopsies have shown promise as diagnostic tools and potential biomarkers for glioblastoma treatment. Plasma samples from glioblastoma patients and healthy individuals can be used to isolate EVs through differential ultracentrifugation, and the content of these EVs can be characterized using mass spectrometry. The cargo of EVs provides insights into the molecular characteristics and subtype information of glioblastoma cells, offering potential applications in treatment strategies. Plasma, serum, urine, and CSF are considered biosources of tumor-associated EVs and biomarkers that reflect the biological status and tumor condition [81,82,83].

EV biomarkers have the potential for diagnostic purposes, disease monitoring, prognostication, and even therapeutic applications for screening and assessing various circulating EV cargos [84]. Different techniques are available for isolating EVs, including ultracentrifugation, density gradient ultracentrifugation, size-based isolation techniques like ultrafiltration and size exclusion chromatography, as well as commercial kits such as ExoQuick^TM^. The choice of isolation technique depends on the specific research question and available resources, as these techniques vary in terms of efficiency and purity [85].

Detecting tumors in glioblastoma patients has been a significant challenge. Studies have shown that EVs derived from the serum of glioblastoma patients exhibit upregulated levels of specific proteins such as C1QA, CD29, CD44, CD81, and CD146, as well as histone H3 compared to EVs from healthy individuals. This distinction in protein expression on the surface of EVs provides a valuable resource for detecting the progression of tumors [86,87].

EVs are a heterogeneous group of lipid-bilayer bound nanoparticles with sizes ranging from 30 nm to 10 μm. They are released by various cell types and carry functional molecules involved in pathological and physiological processes. In the field of oncology, EVs play roles in tumor-microenvironment interactions, cell proliferation, migration, and immunosuppression. Their involvement in intercellular communication is particularly relevant [88,89,90,91].

The content of EVs in biofluids, including subtype-specific markers like CD44, provides valuable information about the phenotype of glioblastoma and can serve as diagnostic and personalized treatment indicators [92,93]. EVs cargo can originate from both blood and CSF, and its unique characteristic of being able to cross the BBB makes it an invaluable transporter of potential biomarkers for glioblastoma [87,94].

### 4.1. Exosomes

Exosomes play a crucial role in intercellular communication by facilitating the transfer of bioactive molecules, including proteins, lipids, and nucleic acids. Their involvement in various pathophysiological processes, including the modulation of glioma-related signaling pathways involved in proliferation, invasion, angiogenesis, immune evasion, and treatment resistance, contributes to the progression of tumors. NcRNAs such as miRNAs and circular RNAs (circRNAs) are part of the exosomal cargo and contribute to the pathophysiology of cancer. These ncRNAs have been shown to modulate glioma-related processes and hold potential as biomarkers (Figure 2) [95,96].

Exosomes are formed through endocytosis at the plasma membrane, leading to the development of multivesicular endosomes (MVEs) and subsequent exosome biogenesis. However, the molecular mechanisms underlying these processes are not yet fully understood [97,98].

Exosomes serve as intercellular messengers, originating from both cancerous and non-cancerous cells, and carrying the molecular characteristics of their parent cells. These small vesicles contain DNA, RNA, and proteins that reflect the cancerous state and can influence various cellular processes, including the proliferation of malignant cells. The presence of tumor-specific DNA sequences in exosomes makes them valuable cancer biomarkers [99]. Importantly, exosomes have the ability to cross the BBB and other anatomical compartments, allowing for the detection of tumor-specific DNA molecules in body fluids. This technique, referred to as “liquid biopsy”, has significant clinical implications for personalized therapies, diagnosis, and prognosis in glioblastoma [11,100]. The molecular composition of exosomes released by glioblastoma cells holds promise for the development of non-invasive detection methods, particularly through the identification of specific protein markers in peripheral blood. A study utilizing two-dimensional electrophoresis (2DE) successfully visualized exosomal proteins from glioblastoma cells, identifying 133 common proteins among glioblastoma samples. Gene ontology analysis further confirmed the presence of these proteins in glioblastoma [101].

The discovery of miRNAs within exosomes has generated significant interest in the field. Extensive studies have demonstrated the specific sorting of miRNAs into exosomes, allowing them to participate in cell-to-cell communication within the tumor microenvironment and exert important roles in tumor biology. Exosomal miRNAs are abundant, stable, and easily accessible in biofluids, making them attractive biomarkers for various cancers, including gliomas [97,102]. Certain exosomal miRNAs, such as hsa-miR-19b-3p, hsa-miR-183–5p, and hsa-miR-323a-3p, have shown promise as noninvasive biomarkers in serum samples for the diagnosis of glioblastoma compared to healthy individuals. These miRNAs directly regulate specific genes and pathways that contribute to the malignant characteristics of glioblastoma cells, suggesting that targeted regulation of these miRNAs could be a potential therapeutic strategy for glioblastoma [103].

Exosomal mRNAs are also recognized as important indicators of cancer, as a result of a tumor cell’s ability to express tumor-specific mRNAs or alter the amounts at which healthy exosomal mRNAs are expressed. These exosomal mRNAs hold potential as biomarkers for assessing drug resistance, a significant challenge in cancer therapy. For example, the levels of MGMT and N-methylpurine DNA glycosylase (*APNG*) mRNA in exosomes derived from glioblastoma patients have been found to be associated with temozolomide resistance levels and treatment effectiveness. These findings highlight the potential of exosomal mRNAs as biomarkers for predicting drug resistance and treatment outcomes in glioblastoma [104,105].

The clinical application of exosomes faces challenges such as large-scale production, standardized isolation techniques, drug loading, stability, and quality control. While significant progress has been made in basic exosome research, therapeutic drug delivery via exosomes still poses challenges [99,106,107,108].

### 4.2. Microvesicles

Microvesicles (MVs) are a diverse population of vesicles that are generated through the process of budding from the cellular membrane. In the context of cancer, it has been observed that cancer cells tend to release a greater number of MVs compared to normal cells [109]. These MVs are formed through membrane blebbing, where components of the cellular membrane, cytoplasmic contents, mRNA, and various proteins from the originating cell are encapsulated within the MVs. This enables the horizontal transfer of these cargo molecules to neighboring cells, facilitating intercellular communication and potentially influencing cellular behavior and functions [110,111]. These freely floating vesicles in the extracellular matrix can be easily obtained from patients’ blood and characterized using standard methods like conventional flow cytometry, which exploits the surface expression of various molecules [112].

The identification of biomarkers, including specific miRNAs and the circulating MVs that transport them, holds significant importance in comprehending the progression, recurrence, and response to treatment in glioblastoma. In a study conducted by Simionescu et al., distinct expression patterns of miRNAs in MVs derived from GBM patients compared to healthy individuals were observed. Furthermore, the study demonstrated that microvesicle parameters decreased following surgical resection of glioblastoma tumors. This finding suggests the potential utility of miR-625-5p as a biomarker for monitoring glioblastoma regression or recurrence [113]. EVs, including exosomes and microvesicles, hold immense potential as diagnostic tools, therapeutic targets, and sources of biomarkers in the field of glioblastoma research. Further exploration of their molecular phenotypes, cargo content, and isolation techniques will contribute to a better understanding of glioblastoma pathogenesis and the development of improved detection and treatment strategies [113].

## 5. Conclusions

The use of biomarkers in body fluids holds promise for the detection, diagnosis, and monitoring of glioblastoma. CTCs, EVs, and cell-free nucleic acids, such as cfDNA and cfRNA, offer valuable insights into the genetic and phenotypic characteristics of GBM. CTCs can provide information on tumor metastasis and treatment response, while cfNAs can reveal genetic and epigenetic alterations specific to GBM. These biomarkers have shown potential as diagnostic and prognostic tools, aiding in personalized treatment strategies and disease management. However, these biomarkers in liquid biopsy have not been widely established or routinely used in clinical practice yet. Several challenges remain in optimizing detection methods, ensuring specificity and sensitivity, and standardizing protocols for isolation and characterization. Further research, validation studies, and prospective clinical trials are crucial to establish the clinical utility of these biomarkers in GBM. These investigations will be instrumental in guiding treatment strategies and ultimately improving patient outcomes, enhancing our understanding of this aggressive brain tumor, and tailoring personalized therapies.

## Figures and Tables

**Figure 1 cancers-15-03804-f001:**
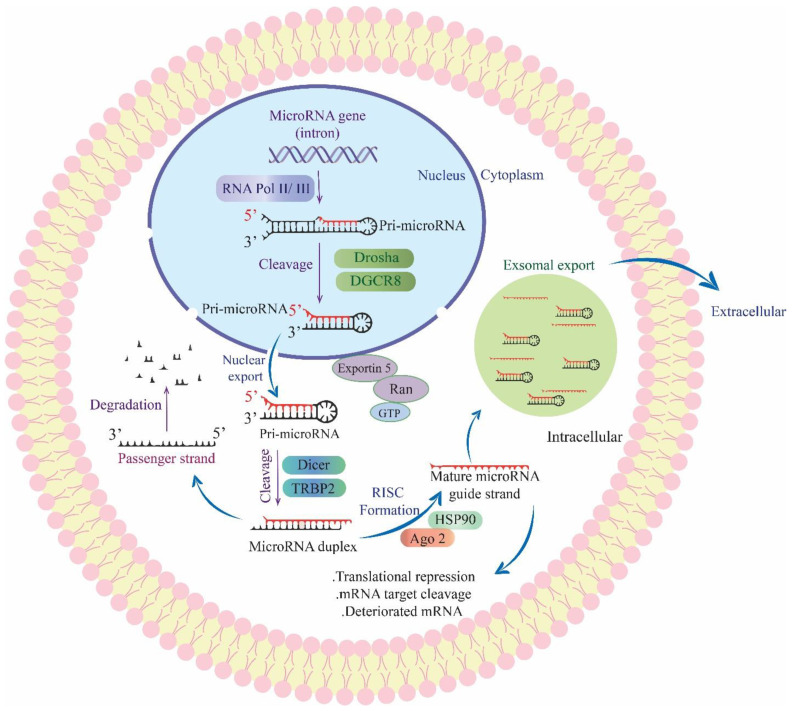
miRNA biogenesis. The biogenesis of miRNAs involves a series of steps. It begins with the transcription of primary miRNA (pri-miRNA) by RNA polymerase II/III. The pri-miRNA is then cleaved by the Drosha-DGCR8 complex, resulting in the formation of pre-miRNA. The next step involves the export of pre-miRNA from the nucleus to the cytoplasm with the assistance of Exportin-5 and Ran-GTP. In the cytoplasm, the Dicer-TRBP complex further processes pre-miRNA, leading to the generation of a miRNA duplex. The duplex is then unwound, resulting in a mature miRNA strand that is loaded onto the Ago2 protein and incorporated into the RNA-induced silencing complex (RISC). The passenger strand of the miRNA duplex is degraded. Finally, the mature miRNA within the RISC can bind to target mRNAs, leading to mRNA cleavage, repression of translation, or degradation of the mRNA. It is worth noting that miRNAs can be selectively released through microvesicles and exosomes.

**Figure 2 cancers-15-03804-f002:**
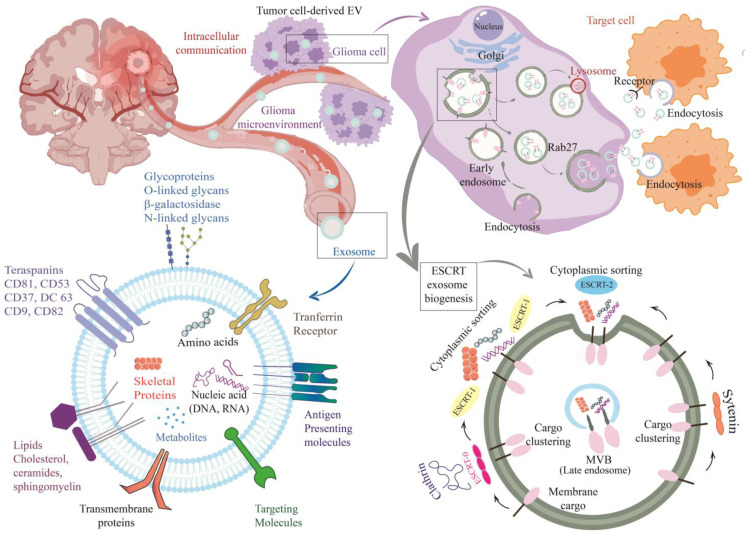
The vital series of actions of biogenesis and release of exosomes. The formation of exosomes involves a series of important steps. It begins with the internalization of the plasma membrane through endocytosis, leading to the formation of early endosomes. These early endosomes mature and transform into multivesicular bodies (MVBs), which contain intraluminal vesicles (ILVs). The formation of ILVs occurs through an endosomal sorting process that can be dependent on the ESCRT (endosomal sorting complex required for transport) machinery or ESCRT-independent pathways. Once the ILVs are generated within MVBs, two fate options arise. The MVBs can either be directed to degradation by fusing with lysosomes, or they can fuse with the plasma membrane, releasing exosomes into the extracellular space. Once released, exosomes can be taken up by target cells through various mechanisms, including endocytosis, fusion with the plasma membrane, or interaction between ligands on the exosome surface and receptors on the target cell surface.

**Table 1 cancers-15-03804-t001:** Ongoing clinical trials using body fluids for GBM detection and monitoring.

Study Title	Status	Biospecimen	Study Type and Time Frame	ClinicalTrials.gov Identifier
Unique Blood and CSF Metabolic Profile Association with Gliomas in Adults	Recruiting	Plasma CSF	Observational 10 years	NCT03865355
A Biospecimen Collection Study in BRAF-V600E Mutated Recurrent Gliomas	Recruiting	Blood CSF Tumor tissue	Observational Procedure: Surgical Cohort 6 years	NCT03593993
Testing Cerebrospinal Fluid for Cell-free Tumor DNA in Children, Adolescents, and Young Adults with Brain Tumors	Not Yet Recruiting	CSF Tumor tissue	Observational 5 years	NCT05934630
Evaluating the Role of Cerebrospinal Fluid (CSF) Cell-free DNA (cfDNA) as a Prognostic Biomarker in Glioblastoma	Recruiting	● CSF ○ ctDNA	Observational Diagnostic Test: Lumber Puncture 5 years	NCT05927610
Longitudinal Assessment of Marrow and Blood in Patients with Glioblastoma	Not Yet Recruiting	Peripheral blood sample collection Tumor collection Bone marrow aspiration	Observational 1 year	NCT04657146
Sonobiopsy for Noninvasive and Sensitive Detection of Glioblastoma	Recruiting	● Blood ○ ctDNA	Interventional 3 years	NCT05281731
Profiling Program of Cancer Patients with Sequential Tumor and Liquid Biopsies	Recruiting	Blood Tumor sample	Interventional Parallel Assignment 4 years	NCT05099068
Evaluating the Expression Levels of MicroRNA-10b in Patients with Gliomas	Recruiting	Blood Tumor tissue CSF	Observational 11 years	NCT01849952

**Table 2 cancers-15-03804-t002:** miRNAs identified in body fluids as GBM biomarkers.

MiRNA	Body Fluid	Regulatory Status	References
miR-21	Blood	Upregulated	[21,63,64,65]
miR-221/222	Blood	Upregulated	[66]
miR-106a-5p, miR-20a-5p, miR-222-3p	Blood	Upregulated	[67]
miR-514a-3p, miR-592	Blood	Upregulated	[68]
miR-340, miR-626, miR-576-5p	Blood	Upregulated	[69]
miR-145-5p	Blood	Downregulated	[70]
miR-125b	Blood	Downregulated	[71]
miR-137	Blood	Downregulated	[72]
miR-203	Blood	Downregulated	[73]
miR-342-3p, miR-628-3p	Blood	Downregulated	[74]
miR-485-3p	Blood	Downregulated	[75]
miR-128	Blood and Plasma	Downregulated in plasma Upregulated in whole blood	[74]
miR-21, miR-10b	CSF	Upregulated	[76,77]
miR-21, miR-15b	CSF	Upregulated	[77]
miR-17, miR-25, miR-23a, miR-27a, miR-106, miR-130	CSF	Upregulated	[77]
miR-193b-3p, miR-331-3p, miR-21-5p, miR-218-5p, miR374a-5p	CSF	Upregulated	[76]
Exosomal miR-21	CSF	Upregulated	[78]
miR-128	CSF	Downregulated	[77]
miR27b-3p, miR-30b-3p, miR548c-3p, miR520f-3p	CSF	Downregulated	[76]

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
