# Peer review of "The Current Landscape of Glioblastoma Biomarkers in Body Fluids"

_cancers, 2023, doi:10.3390/cancers15153804_

Round 1
Reviewer 1 Report
Dear Authors,
Your manuscript on the current landscape of glioblastoma biomarkers in body fluids is a valuable contribution to the field of glioma research. It is well written and very comprehensive. Nevertheless I do have some points that could benefit from some adaptions:
1. In your summary and the introduction you mention that liquid biopsies can be utilized for detection, monitoring and prognosis of GBM. The aspect of detection gets a bit lost in the rest of the manuscript. Are there any studies in which liquid biopsies are used for the initial diagnosis of GBM? Or as it only carried out on patients in which GBM was already detected using other methods?
2. In 2. you mention metastasis quite a lot and repeat this in the conclusion (5. Line 464) again. I think this point is too much in focus due to the fact that GBM virtually do not metastasize. You also mention this in 2., but this does not counterbalance the mentioning of metastasis in the rest of the text.
3. In line 262 you mention IDH mutations to be important for GBM prognosis and prediction, which is a bit misleading. Any tumor with IDH mutation is an astrocytoma or oligodendroglioma and not considered to be a GBM. So it is an important diagnostic marker to distinguish tumor entities, but has no implementation in GBM prognosis and prediction. I would rather consider chromosome 7 gain to be another one.
4. To my impression one aspect of the utilization of DNA and RNA is a bit underrepresented in your manuscript and this is the sequencing of the mentioned fragment. Is this done and if yes, does it give any insight in the tumor progression for example by the detection of new mutations?
Best regards
Author Response
"Please see the attachment."

Reviewer 2 Report
In this manuscript, Zanganeh et al. present a well-constructed review of GBM biomarkers in body fluids including CSF and blood. The authors provided the current landscape and limitations of liquid biomarkers in GBM. This topic will be of great interest to scientists, pathologists, and clinicians in neuro-oncology. However, I would suggest the following points to be included in this manuscript.
1. There should be a summary for approved liquid biopsy test (such as guardant360) in other solid cancers, and why GBM is more challenging compared to other tumors.
2. How’s the consistency of genomic profiling between primary tumors and CTC, ct/cfDNA in GBM?
3. It is important to know IDH, 1p19q status to diagnose glioma. Is there any tests to these genetic status using liquid biopsy? If not, the authors should address this issue as the limitation.
Minor points:
It would be better to use the term just “glioblastoma” as “glioblastoma multiform” is no longer being used.
Author Response
"Please see the attachment."

Reviewer 3 Report
The authors aim to review glioblastoma biomarkers in body fluids to enhance the clinical management of patients with this typically incurable brain tumor. While the manuscript covers all the necessary aspects and is well-structured, with good figures, it does not seem to introduce any significant new aspects that haven't already been included in several recent reviews on the same topic. The conclusion appears overly optimistic, as none of the presented markers seem to have immediate clinical applicability. Moreover, the relevance of these markers to clinical outcomes will likely be negligible due to the limited treatment options available for these tumors. It is recommended to discuss this aspect in more detail to improve the manuscript.
Minor Issues: Line 45: The statement "Grade IV astrocytoma or GBM" is misleading. Please correct it. Grade IV astrocytoma has recently been introduced into the WHO classification of tumors of the nervous system and is clearly distinguished from GBM IV. Line 325/362: The first detection of CD44 variants in glioblastoma was reported by Peter Herrlich's group in the 1990s. This finding may be explored as a potential marker in the future.
Author Response
"Please see the attachment."
